# *CD147* rs8259T>A Variant Confers Susceptibility to COVID-19 Infection within the Mexican Population

**DOI:** 10.3390/microorganisms11081919

**Published:** 2023-07-28

**Authors:** Luis M. Amezcua-Guerra, Carlos A. Guzmán-Martín, Isela Montúfar-Robles, Rashidi Springall, Adrián Hernández-Díazcouder, Rosa Elda Barbosa-Cobos, Fausto Sánchez-Muñoz, Julián Ramírez-Bello

**Affiliations:** 1Immunology Department, Instituto Nacional de Cardiología Ignacio Chávez, Mexico City 14080, Mexico; lmamezcuag@gmail.com (L.M.A.-G.); raspringall@yahoo.com (R.S.); 2Postgraduate Doctoral Program in Biological and Health Sciences, Universidad Autónoma Metropolitana, Mexico City 14387, Mexico; mcarlos93@gmail.com; 3Research Unit, Hospital Juárez de México, Mexico City 07760, Mexico; ismontufar@gmail.com; 4Obesity and Asthma Research Laboratory, Hospital Infantil de México Federico Gómez, Mexico City 06720, Mexico; adrian.hernandez.diazc@hotmail.com; 5Rheumatology Department, Hospital Juárez de México, Mexico City 07760, Mexico; rebcob@yahoo.com; 6The American British Cowdray Medical Center, Mexico City 05348, Mexico; 7Endocrinology Department, Instituto Nacional de Cardiología Ignacio Chávez, Mexico City 14080, Mexico

**Keywords:** CD147, Basigin, single nucleotide polymorphism, rs8259, COVID-19

## Abstract

Background: Coronavirus disease 2019 (COVID-19) is caused by the severe acute respiratory syndrome coronavirus 2 (SARS-CoV-2). Clinical manifestations of COVID-19 range from mild flu-like symptoms to severe respiratory failure. Nowadays, extracellular matrix metalloproteinase inducer (EMMPRIN), also known as cluster of differentiation 147 (CD147) or BASIGIN, has been studied as enabling viral entry and replication within host cells. However, the impact of the *CD147* rs8259T>A single nucleotide variant (SNV) on SARS-CoV-2 susceptibility remains poorly investigated. Objective: To investigate the impact of rs8259T>A on the *CD147* gene in individuals from Mexico with COVID-19 disease. Methods: We genotyped the *CD147* rs8359T>A SNV in 195 patients with COVID-19 and 185 healthy controls from Mexico. In addition, we also measured the expression levels of CD147 and TNF mRNA and miR-492 from whole blood of patients with COVID-19 through RT-q-PCR. Results: We observed a significant association between the *CD147* rs8259T>A SNV and susceptibility to COVID-19: T vs. A; OR 1.36, 95% CI 1.02–1.81; *p* = 0.037; and TT vs. AA; OR 1.77, 95% CI 1.01–3.09; *p* = 0.046. On the other hand, we did not find differences in CD147, TNF or miR-492 expression levels when considering the genotypes of the *CD147* rs8259T>A SNV. Conclusions: Our results suggest that the *CD147* rs8259T>A variant is a risk factor for COVID-19.

## 1. Introduction

Coronavirus disease 2019 (COVID-19), caused by the severe acute respiratory syndrome coronavirus 2 (SARS-CoV-2), has resulted in over 6.8 million deaths worldwide [1,2]. The clinical manifestations of COVID-19 range from mild flu-like symptoms to severe respiratory failure [3]. The ability of SARS-CoV-2 to infect host cells is primarily governed by the interactions between the viral spike (S) protein and its human receptor angiotensin-converting enzyme-2 (ACE2). These interactions initiate a cascade of events that enable viral entry and replication within the host cells [4]. Initially, the human ACE2 receptor was identified as the primary host cell receptor for the S protein [5]. Nevertheless, further investigations revealed additional molecules involved in mediating viral infection, such as extracellular matrix metalloproteinase inducer (EMMPRIN), also known as cluster of differentiation 147 (CD147) or BASIGIN [6]. Although the impact of CD147 on SARS-CoV-2 infection remains poorly investigated, its significance should not be underestimated.

CD147, a highly glycosylated transmembrane glycoprotein belonging to the immunoglobulin superfamily, is expressed in diverse immune cells and plays a role in triggering inflammation with the release of cytokines such as interleukin (IL)-6 and tumor necrosis factor (TNF), induction of matrix metalloproteinase 2 (MMP)-2, IL-9, interferon (IFN)-gamma, and immune cell activation [7]. In recent years, genetic research has been focused on understanding how single nucleotide variants (SNVs) can influence infectious disease susceptibility. SNVs are among the most common genetic variations, occurring approximately every 1200 base pairs (bp) when comparing a pair of human chromosomes [8]. This information is often used to uncover why some people are more likely to have a certain disease or condition than others, including COVID-19.

For instance, a meta-analysis conducted by Li and colleagues assessed the association between interferon-induced transmembrane protein 3 (*IFITM3*) variants and susceptibility to COVID-19. Notably, they identified an association between the *IFITM3* rs12252A>G SNV and COVID-19 infection [9]. On the other hand, some studies on SNVs in COVID-19 have been focused on genes encoding SARS-CoV-2 receptors or on proteases that prime the S protein, such as ACE2 and TMPRSS2, respectively [10,11]. In addition, a recent study evaluated the *CD147* rs8259T>A SNV in patients with COVID-19 [12]. CD147 is another receptor of the S protein [13]. Izmailova and colleagues aimed to investigate potential associations between the *CD147* rs8259T>A, *ACE2* rs4240157T>C, *TMPRSS2* rs12329760C>T, and *TMPRSS11A* rs353163C/T variants and COVID-19 severity in the Ukrainian population. They analyzed these variants among cases divided into three groups: without oxygen therapy, non-invasive oxygen therapy, and invasive oxygen therapy. Interestingly, they only found frequency differences for the *TMPRSS2* rs12329760C>T variant in the group receiving invasive oxygen therapy, but they did not identify an association between *CD147* rs8259T>A and COVID-19 susceptibility or severity [12]. However, given the sample size reported in that study, it is necessary to evaluate whether this same variant is a risk factor in populations with a different genetic background. The *CD147* rs8259T>A variant is in the 3′-untranslated region (3′ UTR) of *CD147* on chromosome 19:582927 and has been evolutionarily conserved in several eutherian mammals (Figure 1a–c).

Despite several studies investigating different SNVs in genes such as *TMPRSS2*, *ACE1*, *ACE2*, and others with COVID-19 susceptibility, it remains unclear whether the *CD147* rs8259T>A variant confers susceptibility to this infectious disease in the Mexican population. Therefore, our objective was to investigate the impact of the *CD147* rs8259T>A variant in individuals from Mexico who have contracted COVID-19. In addition, we evaluated whether the genotypes of the *CD147* variant are associated with CD147, TNF, and miR-492 expression levels. Thus, we aim to contribute to the understanding of the genetic factors influencing COVID-19 susceptibility.

## 2. Materials and Methods

### 2.1. Study Population

In this study, a total of 195 symptomatic patients with COVID-19 confirmed by RT-qPCR test were enrolled from the Hospital Juárez de México. The recruitment period spanned from October 2020 and January 2021, during which the original strain of SARS-CoV-2 was observed. The diagnosis of COVID-19 was made based on clinical characteristics, such as loss of taste, dry cough, fatigue, fever, diarrhea, nasal congestion, sore throat, conjunctivitis, headache, musculoskeletal pain, skin rashes, dizziness, heart rate, and oxygen saturation, together with a positive PCR test for SARS-CoV-2. Laboratory parameters were collected from medical records on the same day as the study sampling. For the control group, we included 195 individuals who were not infected with SARS-CoV-2, and their samples were collected during the period of 2016 to 2018. Participants in the control group had no history of chronic-inflammatory, autoimmune, or cancer diseases (conditions where CD147 is widely expressed), and they were free from respiratory diseases. Additionally, individuals in the control group tested negative for COVID-19 infection based on RT-qPCR. Our study was conducted in accordance with the principles outlined in the Declaration of Helsinki and received approval from the Ethics, Biosecurity, and Research Committees of the Hospital Juárez de México (project number HJM 024/22-I). All patients or their relatives provided informed consent by signing the institutional consent letter.

### 2.2. Genetic Analysis

Whole blood samples (5 mL) were collected in EDTA tubes following WHO recommendations for sampling and processing biological samples from COVID-19 cases [14]. Genomic DNA was extracted using the QIAamp DNA Blood Mini kit (QIAGEN, Hilden, Germany). DNA integrity was evaluated by agarose gel electrophoresis (1% agarose gel stained with ethidium bromide). The purity and concentration of the extracted DNA were measured using a NanoDrop 2000 spectrophotometer (Thermo Fisher’s Scientific, Wilmington, DE, USA). Genotyping of the rs8259T>A variant was performed using Applied Biosystems TaqMan Genotyping Assays (Foster City, CA, USA), according to the manufacturer’s instructions.

### 2.3. CD147, miR-492, and TNF mRNA Expression

K-EDTA whole blood samples (400 μL) were isolated using Tripure reagent and subsequently treated with DNAse I (Roche, Penzberg, Germany). For CD147, TNF, and GAPDH mRNA levels, 50 ng of total RNA was amplified by one-step RT-qPCR with the QuantiNova Probe RT-PCR Kit (Qiagen, Hilden, Germany).

For miR-492 and U6 (used as endogenous control) levels, 20 ng of total RNA was retrotranscribed using the TaqMan miRNA RT kit (Applied Biosystems, Foster City, CA, USA). For qPCR, 1 μL of cDNA was used along with the QuantiNova Probe PCR Master Mix (Qiagen), following the manufacturer’s instructions.

CD147 mRNA levels were compared with whole blood from 25 patients and 25 controls. In addition, we evaluated the levels of miR-492, CD147, and TNF mRNA considering the three genotypes of *CD147* rs8259T>A from 25 randomized samples of patients with COVID-19, maintaining the same percentages of genotypes identified in all patients with this infectious disease.

The qPCR was performed using the Oppus CFX96 system (BioRad, Hercules, CA, USA). The cycling conditions consisted of an initial denaturation at 95 °C for 10 min, followed by 45 cycles at 95 °C for 15 s, 60 °C for 60 s, and 72 °C for 1 s. Expression levels were measured in duplicate and normalized using glyceraldehyde-3-phosphate dehydrogenase (GAPDH) sequence 5′-AGCCACATCGCTCAGACAC-3′ and 5′-GCCCAATACGACCAAATCC-3′ as the reference. The following NCBI assay genes and TaqMan catalog numbers were used: CD147: Hs00936295_m1, Catalog #4331182; TNF: Hs00174128_m1, Catalog #4331182; hsa-miR-492: 001039, U6: Catalog #4427975 (Thermo Fisher, Foster City, California). Relative quantification was performed using the 2 −delta Ct (2^−∆Ct^) method.

### 2.4. Statistical Analysis

Hardy-Weinberg equilibrium analysis was conducted using the web software https://ihg.helmholtz-muenchen.de accessed on 17 January 2023. Alleles and genotype frequencies were calculated using GraphPad Prism software v8. To evaluate susceptibility between *CD147* variants and COVID-19, we compared allelic and genotypic frequencies between SARS-CoV-2-infected patients and controls. Additional statistical analyses were carried out in SPSS v26. A comparison of expression levels of TNF, miR-492, and CD147 mRNAs between different genotype groups was carried out using the Kruskal–Wallis test. The association between genotypes and susceptibility to COVID-19 infection was evaluated using the chi-square test. Correlation analysis was performed using the Pearson test. A *p*-value less than 0.05 was considered statistically significant.

## 3. Results

Among the 195 COVID-19 patients included in our study, a median age of 55 years (IQR 46 to 66) was observed, and 122 (62.6%) patients were male. Mechanical ventilation was required for 60 (30.8%) patients during hospitalization. Clinical improvement was observed in 68.2% of the cases, while 31.8% ultimately died. The main clinical and laboratory data are summarized in Table 1.

Regarding the allele and genotypic frequencies in controls, we identified a normal distribution of genotypes of the *CD147* rs8259T>A variant in controls. The distribution of allele and genotype frequencies of the *CD147* rs8259T>A variant in cases and controls is presented in Figure 2. Our analyses revealed, after adjusting for age, gender, comorbidities, etc., an association between the *CD147* rs8259T>A SNV and COVID-19 infection; T vs. A (OR 1.36, 95% CI 1.02 to 1.81, and *p* = 0.037) and TT vs. AA (OR 1.77, 95% CI 1.01 to 3.09 and *p* = 0.046) (Figure 2). However, no significant differences were found in the quantitative laboratory variables or clinical traits associated with COVID-19 among the genotype groups of *CD147* rs8259T>A (Table 2). Our data suggest that this variant is a risk factor for COVID-19, but it is not associated with severity.

Additionally, we compared the expression levels of CD147 mRNA levels in 25 COVID-19 patients and 25 controls, and we found a higher expression in COVID-19 patients compared with controls (COVID-19 median: 2.01, IQR 0.63 to 3.97; controls median 0.96, IQR 0.69 to 1.52, *p* = 0.04) (Figure 3a). Next, we conducted an analysis of the expression of the CD147 (Figure 3b), miR-492 (Figure 3c), and TNF mRNAs (Figure 3d) considering the genotypes of rs8259T/A. After analysis, we did not identify any statistically significant difference between the genotypes of the aforementioned molecules in patients with COVID-19: CD147 (TT genotype median 1.93, IQR 0.697 to 3.32 vs. TA genotype median 2.13, IQR 1.142 to 5.863 vs. AA genotype median 3.02, IQR 0.85 to 11.93; *p* > 0.05; Figure 3b), miR-492 (TT genotype median 0.034, IQR 0.019 to 0.086 vs. TA genotype median 0.028, IQR 0.014 to 0.052 vs. AA genotype median 0.019, IQR 0.007 to 0.076; *p* > 0.05; Figure 3c), and TNF (TT genotype median 4.30, IQR 0.50 to 12.43 vs. TA genotype median 1.87, IQR 0.57 to 3.04 vs. AA genotype median 1.92, IQR 0.70 to 3.32; *p* > 0.05; Figure 3d).

## 4. Discussion

CD147, a transmembrane glycoprotein belonging to the immunoglobulin superfamily, has been implicated in viral infections. Like the human ACE2 receptor, CD147 has been reported to be another human receptor for the S protein of SARS-CoV-2 and to be involved in viral entry [10,15]. In addition, previous studies have demonstrated the functional role of CD147 in facilitating SARS-CoV-2 infection [16]. Indeed, CD147 interacts with the S protein of SARS-CoV-2, thereby participating in the entry route of SARS-CoV-2 into infected human cells [15].

Some authors have suggested that certain variants in the *CD147* gene could be important to the susceptibility and/or severity of COVID-19 [6,17,18], given the significance of CD147 in the entry of SARS-CoV-2 into human host cells [17,18]. Thus, in this study, we investigated whether the *CD147* rs8259T>A variant is associated with COVID-19 infection in the Mexican population. To the best of our knowledge, only one study has previously reported on the role of the rs8259T>A variant in COVID-19 susceptibility and severity in the Ukrainian population; however, they did not find any association [12]. In contrast, our study revealed a significant association with COVID-19 susceptibility. These discrepant findings between the two studies may be attributed to several factors. Firstly, the difference in the number of control samples could have influenced the results. Our study had a larger control group, consisting of 185 Mexican mestizos, compared to their study’s 92 controls. A larger control group provides more statistical power to detect associations accurately. Additionally, the genetic ancestry of the study populations might have played a crucial role. Our study included participants from Central Mexico, where the genetic background is characterized by approximately 52% European, 44% Amerindian, and 4% African ancestry [19]. Indeed, the genotype frequencies of the CD147 rs8259T>A variant in our Mexican controls are different from those identified in the Ukrainian controls [12]. These differences could lead to the fact that in Mexicans it is a risk factor, while in Ukrainians it is not. Given the lack of association reported in the European population and the association we identified in a Latin American population, other studies should be conducted in other populations with different genetic backgrounds to determine whether this CD147 variant is indeed a risk factor for SARS-CoV-2 infection and COVID-19. On the other hand, in our study, we assessed whether the rs8259T>A variant is associated with specific laboratory parameters or clinical features related to COVID-19. However, after analysis, we did not observe any significant associations, indicating that it is not associated with disease severity in our study population.

It is worth noting that the A allele or AA genotype of the *CD147* rs8259T>A variant, which we identified as a risk factor for COVID-19, has previously been reported as a risk factor for chronic heart failure (CHF) in Chinese patients [20]. Moreover, a study by Yingzhen Weng et al. explored the role of SNVs in *CD147* and matrix metalloproteinase-9 (MMP-9) in the susceptibility and severity of coronary artery disease (CAD). They genotyped the rs8259T>A, rs28915400G>T, rs4919859G>C, rs6758G>A, and rs8637G>A *CD147* variants and rs3918242C>T *MMP-9* variant in 812 patients and 258 controls and found associations between rs8259T>A and rs3918242C>T variants and CAD, suggesting their involvement in the pathological process of the disease [21].

Finally, we investigated the expression levels of CD147 mRNA between patients with COVID-19 and controls, as well as whether the three genotypes of rs8259T>A were associated with abnormal levels of CD147, miR-492, and TNF mRNA expression in patients with COVID-19. We observed significantly higher expression levels of CD147 mRNA in patients with COVID-19 compared to controls. This finding suggests that CD147 may play a role in the pathogenesis or progression of COVID-19. The elevated expression of CD147 in patients with COVID-19 is in line with previous studies that have reported an increase of CD147 mRNA in some inflammatory and autoimmune diseases [13,22,23]. These results highlight the potential of CD147 as a biomarker or therapeutic target for COVID-19. The CD147 mRNA or protein is expressed in high levels in whole blood and in PBMCs of controls, as previously reported [6,24,25] and as we also identified in our study. Our analysis did not reveal any statistically significant difference in CD147, miR-492, and TNF expression levels among the three genotypes of rs8259T>A in patients with COVID-19. We did not carry out any study of the expression of CD147, miR-492, and TNF considering the genotypes of the *CD147* rs8259T>A variant in controls, this being one of the limitations of our study.

On the other hand, the A/A and T/A genotypes of the *CD147* rs8259T>A variant have been associated with higher expression levels of CD147 mRNA or protein in PBMCs from patients with ACS but not in patients with stable angina [24]. Thus, the effect of genotypes of the CD147 rs8259T>A variant on the expression of its mRNA appears to depend on the cell type, tissue, or disease. As far as we know, the expression of miR-492 had not been evaluated in patients with COVID-19, but a study showed that this miRNA is expressed in PBMCs from healthy controls [25]. In this sense, we also found an expression of this miRNA in whole blood cells of our controls. miR-492 has been reported to bind to the T allele of the *CD147* rs8259T>A variant; meanwhile, the A allele destroys the binding site for this miRNA, which generates an increase in CD147 protein expression [24]. In this same study, it was reported that the AA genotype of rs8259T>A is associated with an increased production of CD147 protein (but not mRNA) in PBMCs from patients with psoriasis versus the TT genotype [25]. Therefore, the effect of the three genotypes (T/T, T/A, and A/A) of this variant on its protein or mRNA expression levels appears to depend on the cell type, tissue, or disease [24,25]. In line with the information reported by Wu et al., we do not observe statistically significant differences in CD147 mRNA levels considering the genotypes of this variant in patients with psoriasis and stable angina [24,25]. Although we did not identify differences in TNF mRNA expression considering the three rs8259T>A genotypes, we know that TNF is a cytokine that mediates inflammation and can cause detrimental tissue damage, that it promotes lung fibrosis, which later results in pneumonia, pulmonary edema, and acute respiratory distress syndrome, that it promotes inflammation and that it is associated with morbidity and mortality in patients with COVID-19 [26].

It is important to acknowledge the limitations of our study. Although we conducted a comprehensive analysis of the *CD147* rs8259T>A variant in relation to COVID-19, laboratory parameters, and gene expression, our findings are based on a specific population and may not be generalizable to other ethnicities or regions.

## 5. Conclusions

Our data suggest that this *CD147* rs8259T>A variant is a risk factor for COVID-19 in the Mexican population. These findings highlight the potential involvement of CD147 in the pathogenesis of COVID-19 and suggest the importance of genetic factors in disease susceptibility. Understanding the role of CD147 and its variants in COVID-19 may contribute to the development of personalized medicine and targeted therapeutic interventions.

## Figures and Tables

**Figure 1 microorganisms-11-01919-f001:**
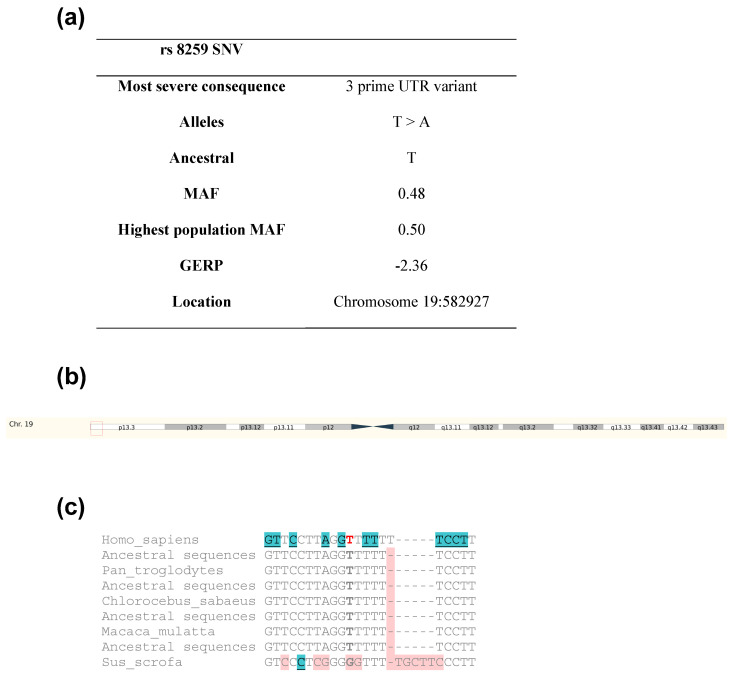
Information on *CD147* rs8259 SNV. (**a**) Some features of CD147 rs8259T>A, (**b**) 19:582927 chromosomic localization (enclosed in the red box), and (**c**) conservation through eutherian mammals. Blue color denotes variants in the 3’ UTR (untranslated region), red indicates the focus variant, and pink represents those that differ from the primary species.

**Figure 2 microorganisms-11-01919-f002:**
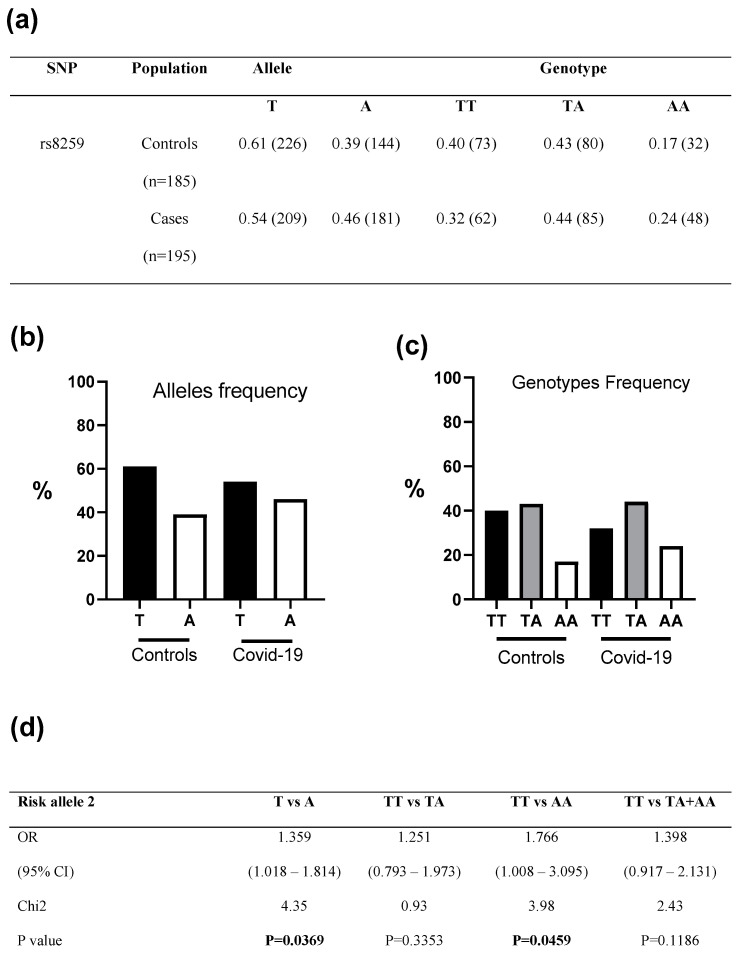
(**a**–**c**) Allele and genotype frequencies of *CD147* rs8259T>A observed on both controls and COVID-19 patients. (**d**) Association analysis of the *CD147* rs8259T>A variant in cases vs. controls CI—Confidence interval, OR—Odds ratio; significant *p*-values are in bold.

**Figure 3 microorganisms-11-01919-f003:**
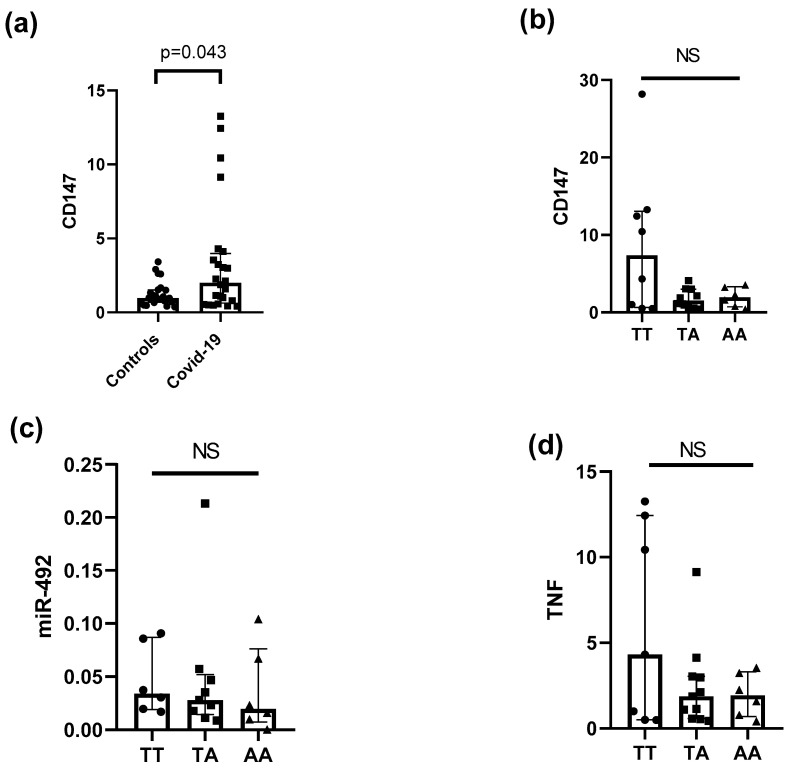
CD147, TNF, and miR-492 mRNA levels in patients with COVID-19. (**a**) CD147 mRNA levels in 25 COVID-19 patients and 25 controls. (**b**) CD147, (**c**) miR-492, and (**d**) TNF mRNA levels considering the three genotypes (TT, TA, and AA) of the *CD147* rs8259T>A in 25 patients with COVID-19. NS: Not Significative.

**Table 1 microorganisms-11-01919-t001:** Demographic and clinical data of COVID-19 participants.

	COVID-19 Patients (*n* = 195)
Age in years, median (IQR)	55 (46–66)
Male sex, *n* (%)	122 (62.6)
Outcome	
Clinical improvement, *n* (%)	133 (68.2)
Death, n (%)	62 (31.8)
Mechanical Ventilation, *n* (%)	60 (30.8)
Diabetes Mellitus, *n* (%)	69 (35.4)
Systemic Hypertension, *n* (%)	82 (42.1)
Obesity, *n* (%)	82 (42.1)
Serum creatinine, median (IQR)	0.87 (0.66–1.35)
Ferritin, median (IQR)	686 (386–1028.7)
Lactic Dehydrogenase, median (IQR)	363.7 (268–471)
C-reactive protein, median (IQR)	8.9 (3.8–23.4)
Total bilirubin, median (IQR)	0.6 (0.43–0.80)
ALT, median (IQR)	47.5 (28–72)
AST, median (IQR)	40 (27–67)

Main clinical characteristics of COVID-19 patients; quantitative variables are represented with median and interquartile range, and qualitative variables are shown with frequencies and percentages. ALT—alanine aminotransferase, AST—aspartate aminotransferase.

**Table 2 microorganisms-11-01919-t002:** Comparison of quantitative and qualitative variables in different genotype groups in patients with COVID-19.

	TT (*n* = 62)	TA (*n* = 85)	AA (*n* = 48)	*p*
Mechanical Ventilation (*n*, %)	18 (29.0)	27 (31.8)	15 (31.3)	0.787
Diabetes mellitus (*n*, %)	24 (38.7)	27 (31.8)	18 (37.5)	0.644
Systemic Hypertension (*n*, %)	25 (40.3)	37 (43.5)	19 (39.6)	0.882
Male (*n*, %)	39 (62.9)	54 (63.5)	29 (60.4)	0.936
Obesity (*n*, %)	29 (46.8)	34 (40)	19 (35.6)	0.659
Creatinine (mg/dL)	0.84 (0.69–1.28)	0.88 (0.62–1.32)	0.96 (0.73–1.42)	0.31
Ferritin (ng/mL)	681 (348–1074)	686 (377–1295)	693 (396–992)	0.78
LDH (IU/L)	366 (256–489)	362 (275–460)	350 (269–558)	0.84
CRP (mg/dL)	6.38 (3.3–21.1)	10.20 (4.0–22.2)	8.32 (4.3–65.2)	0.60

## Data Availability

Raw data are available directly from the corresponding authors with a reasonable request.

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
