# Peer review of "CD147 rs8259T>A Variant Confers Susceptibility to COVID-19 Infection within the Mexican Population"

_microorganisms, 2023, doi:10.3390/microorganisms11081919_

Round 1

Reviewer 1 Report

Amezcua-Guerra et al present a  study on the potential effect of a specific CD147 on the susceptibility to COVID-19. CD147 genotyping using TaqMan assays as well as gene expression assessed by RT-qPCR were performed. The authors conclude that the CD147 rs8259 polymorphism is associated with COVID-19 susceptibility.

The study is very restricted and short. The authors should related it with other genetic studies, in particular with the study by Izmailova et al. (Ref. 11) and should include a more extensive comparative discussion. Also, as CD147 is a host-factor for SARS-CoV-2 and minds to the viral spike protein (Ref. 14) the authors should also discuss the dependence of susceptibility on different viral strains. However, that may exceed the scope of this short paper. Thus, the authors should confirm the specific SARS-CoV-2 strains that were prevalent in their population.

Furthermore, there are concerns regarding the experimental design. Amezcua-Guerra et al report that their cohort consists of 195 symptomatic COVID-19 patients and 185 healthy donors. For the genetic analysis out of 195 symptomatic cases 12 COVID-19 cases were selected? How? What about the 185 healthy cases? The authors should discuss in great detail how they have defined susceptibility and how they have calculated the association between susceptibility and the genotypes by precisely describing how many samples from COVID-19 patients and healthy controls have been used and how they have been selected. With respect to the expression analysis, the authors select 25 randomized samples "of the same patients, maintaining the same percentages of the genotypes analysis." How does 12 samples for the genetic analysis and 25 for the expression analysis maintain the same percentage? What about the expression analysis of the healthy controls? Have this been performed? If not, why not? There could be a natural baseline expression of CD147 in the healthy population which should be compared with the COVID-19 patients.

Figure 3 is not referred to in the main text, which should be added (lines .
TNF and miR-492 expression have been both measured by RT-qPCR, allele-specific expression at least from miR-492 are shown in Fig. 3 but neither TNF nor miR-492 expression is being discussed in the text.

Overall this study leaves many unanswered questions.

Author Response

 Reviewer 1

Amezcua-Guerra et al present a study on the potential effect of a specific CD147 on the susceptibility to COVID-19. CD147 genotyping using TaqMan assays as well as gene expression assessed by RT-qPCR were performed. The authors conclude that the CD147 rs8259 polymorphism is associated with COVID-19 susceptibility.

Question 1. The study is very restricted and short. The authors should relate it with other genetic studies, in particular with the study by Izmailova et al. (Ref. 11) and should include a more extensive comparative discussion. Also, as CD147 is a host factor for SARS-CoV-2 and minds to the viral spike protein (Ref. 14) the authors should also discuss the dependence of susceptibility on different viral strains. However, that may exceed the scope of this short paper. Thus, the authors should confirm the specific SARS-CoV-2 strains that were prevalent in their population.

Answer 1. Dear reviewer, we agree with you. Therefore, we have expanded and give details in our manuscript about other studies comparing their results with ours. Thus, we have added different sentences in our manuscript.

In the discussion section, we have now added more information on CD147 as a receptor of SARS-CoV-2 (paragraph 1, lines 216 – 221).  The following sentence has been added:

Like the human ACE2 receptor, CD147 has been reported to be another human receptor for the S protein of SARS-CoV-2 and be involved in the viral entry (10, 15). In addition, previous studies have demonstrated the functional role of CD147 in facilitating SARS-CoV-2 infection (13). Indeed, CD147 interacts with the S protein of SARS-CoV-2 and thereby participating in the entry route of the SARS-CoV-2 into infected human cells (13, 15).

We have also added more information about other studies, including that of Izmailova et al (introduction section, paragraph 2, lines 59 – 73)

On the other hand, some studies on SNVs in COVID-19 have been focused on genes encoding SARS-CoV-2 receptors or on proteases that prime the S protein, such as ACE2 and TMPRSS2, respectively (10, 11). In addition, a recent study evaluated the CD147 rs8259T>A SNV in patients with COVID-19 (12). CD147 is another receptor of the S protein (13). Izmailova and colleagues aimed to investigate potential associations between the CD147 rs8259T>A, ACE2 rs4240157T>C, TMPRSS2 rs12329760C>T, and TMPRSS11A rs353163C/T variants and COVID-19 severity in the Ukrainian population. They analyzed these variants among cases divided into three groups: without oxygen therapy, non-invasive oxygen therapy, and invasive oxygen therapy. Interestingly, they found frequency differences for the TMPRSS2 rs12329760C>T variant only in the group receiving invasive oxygen therapy but they did not identify an association between CD147 rs8259T>A and COVID-19 susceptibility or severity (12). However, given the sample size reported in that study, it is necessary to evaluate whether this same variant is a risk factor in populations with different genetic background.

We have also added more information in the discussion section (paragraph 2, lines 226 - 244)

To our knowledge, only one study has reported the role of the rs8259T>A variant in the susceptibility and severity of COVID-19 in the Ukrainian population, but it was not associated with COVID-19 susceptibility or severity (12). Our findings differ from those reported in the Ukrainian population because we identified an association with COVID-19 susceptibility, meanwhile, they did not report any association. Some differences between the two studies might explain this discrepancy. Firstly, we had a larger number of control samples, including 185 Mexican mestizos compared to their 92 controls. Ancestry also plays a crucial role, our study includes participants from Central Mexico, which is characterized by approximately 52% European, 44% Amerindian, and 4% African ancestry (20). In fact, genotype frequencies of TT, TA, and AA of the CD147 rs8259T>A variant in our Mexican controls is different from those identified in Ukrainian controls (12). These differences could lead to the fact that in Mexicans it is a risk fac-tor, while in Ukrainians no. Other studies (considering the lack of association reported in the European population and the association we identified in a Latin American population) should be conducted in other populations with different genetic background to determine whether this CD147 variant is indeed a risk factor for SARS-CoV-2 infection and COVID-19.

Question 2. Furthermore, there are concerns regarding the experimental design. Amezcua-Guerra et al report that their cohort consists of 195 symptomatic COVID-19 patients and 185 healthy donors. For the genetic analysis out of 195 symptomatic cases, 12 COVID-19 cases were selected? How? What about the 185 healthy cases? The authors should discuss in great detail how they have defined susceptibility and how they have calculated the association between susceptibility and the genotypes by precisely describing how many samples from COVID-19 patients and healthy controls have been used and how they have been selected. With respect to the expression analysis, the authors select 25 randomized samples "of the same patients, maintaining the same percentages of the genotypes analysis." How does 12 samples for the genetic analysis and 25 for the expression analysis maintain the same percentage? What about the expression analysis of the healthy controls? Have this been performed? If not, why not? There could be a natural baseline expression of CD147 in the healthy population which should be compared with the COVID-19 patients.

Answer 2. We have added some sentences to clarify each point you have mentioned.

In the material and methods section, we have added the following sentence to clarify important points about the selection of cases and controls (lines 88 – 104).

The recruitment period spanned from October 2020 and January 2021, where the Original strain variant was observed. The diagnosis of COVID-19 was made based on clinical characteristics such as loss of taste, dry cough, fatigue, fever, diarrhea, nasal congestion, sore throat, conjunctivitis, headache, musculoskeletal pain, skin rashes, dizziness, heart rate, and oxygen saturation, together with a positive PCR test for SARS-CoV-2. Laboratory parameters were collected from medical records on the same day as the study sampling. Our control group included 195 individuals not infected with SARS-CoV-2 (collected during 2016 - 2018) and without a history of chronic inflammatory, autoimmune, and cancer diseases (diseases where CD147 is widely expressed) and free from respiratory diseases and negative for COVID-19 infection based on RT-qPCR. Our study complied with the Declaration of Helsinki and was approved by the Ethics, Biosecurity, and Research Committees from the Hospital Juárez de México (project number HJM 024/22-I). All patients or their relatives signed the institutional consent letter.

To clarify susceptibility, we have added the following sentence in the statistical analysis section (lines 143 - 144).

To evaluate susceptibility between CD147 variants and COVID-19, we compared allelic and genotypic frequencies between SARS-CoV-2-infected patients and controls

We have also added the following sentence to answer about gene expression in material and method section (subheader CD147, miR-492, and TNF mRNA expression, paragraph 3, lines 125 - 129)

CD147 mRNA levels were compared of whole blood from 25 patients and 25 controls. In addition, we evaluated the levels of miR-492, CD147, and TNF mRNA taking into account the three genotypes of CD147 rs8259T>A from 25 randomized samples of patients with COVID-19, maintaining the same percentages of genotypes identified in all patients with this infectious disease.

It is important to mention that we initially recruited 40 patients with COVID-19 to carry out the expression analysis of CD147, miR-492, and TNF, of these 40 individuals, we randomly selected 25 of them who represented the same percentage of genotypes as we identified in the 195 patients who were included in our study. We have respectfully decided not to include any sentence in the manuscript regarding this clarification.

Regarding the analysis of expression in controls, we have added various sentences in different sections of our manuscript, for example, in the material and method section (subheader CD147, miR-492, and TNF mRNA expression, paragraph 3, line 125) we added the following sentence:

CD147 mRNA levels were compared of whole blood from 25 patients and 25 controls.

In addition, we have also added the following sentence in results section (paragraph 3, lines 170 – 173).

Additionally, we compared the expression levels of CD147 mRNA levels in 25 COVID-19 patients and 25 controls, we found higher expression in COVID-19 patients com-pared with controls (COVID-19, median: 2.01, IQR 0.63 to 3.97, controls median 0.96, IQR 0.69 to 1.52, p=0.04) (Figure 3a).

In Figure 3a, we have added the following sentence:

CD147 mRNA levels in 25 COVID-19 patients and 25 controls (3a)

In discussion section, paragraph 4, lines 258 - 261, we have added the following phrase:

Finally, we investigated the expression levels of CD147 mRNA between patients with COVID-19 and controls.

In this same paragraph 3, lines 267 - 269 of discussion section, we have added the following sentence:

The CD147 mRNA or protein is expressed in high levels in whole blood and in PBMCs of controls as previously reported (6,23,24) and as we also identified in our study.

Finally, we have added the following sentence in this same paragraph 3 (discussion section), lines 271 – 273.

We did not carry out any study of expression of CD147, miR-492, and TNF considering the genotypes of the CD147 rs8259T>A variant in controls, being one of the limitations of our study.

In discussion section (paragraph …, lines ….), we have added information explaining why controls were not included in the gene expression analysis and therefore the three CD147 rs8259T>A genotypes in healthy individuals were not taken into account. Thus, we have added the following sentence:

Question 3. Figure 3 is not referred to in the main text, which should be added (lines...
TNF and miR-492 expression have been both measured by RT-qPCR, allele-specific expression at least from miR-492 are shown in Fig. 3 but neither TNF nor miR-492 expression is being discussed in the text.

Overall this study leaves many unanswered questions.

Answer 3. Now we have added the reference to Figure 3 (a, b,c and d) in our manuscript. In addition, we have included additional information about the expression of miR-492 and TNF in our manuscript. Thus, we have added the following sentence in the results section (paragraph 3, lines 173 - 182):

Next, we conducted an analysis of the expression of the CD147 (3b), miR-492 (3c), and TNF mRNAs (3d) taking into account the genotypes of rs8259T/A. After analysis, we did not identify any statistically significant difference between the genotypes of the aforementioned molecules in patients with COVID-19: CD147 (TT genotype median 1.93, IQR 0.697 to 3.32 vs. TA genotype median 2.13, IQR 1.142 to 5.863 vs. AA genotype median 3.02, IQR 0.85 to 11.93; p >0.05; Figure 3b), miR-492 (TT genotype median 0.034, IQR 0.019 to 0.086 vs. TA genotype median 0.028, IQR 0.014 to 0.052 vs. AA genotype median 0.019, IQR 0.007 to 0.076; p >0.05; Figure 3c), and TNF (TT genotype median 4.30, IQR 0.50 to 12.43 vs. TA genotype median 1.87, IQR 0.57 to 3.04 vs. AA genotype median 1.92, IQR 0.70 to 3.32; p >0.05; Figure 3d).

We have also added the following information in the discussion section (paragraph 4, lines 259 - 260):

and whether the three genotypes of rs8259T>A were associated with abnormal levels of CD147, miR-492, and TNF mRNAs expression in patients with COVID-19.

In addition, we have added the following sentence in discussion section, paragraph 4, lines 269 – 273.

Our analysis did not reveal any statistically significant difference in CD147, miR-492, and TNF expression levels among the three genotypes of rs8259T>A in patients with COVID-19. We did not carry out any study of expression of CD147, miR-492, and TNF considering the genotypes of the CD147 rs8259T>A variant in controls, being one of the limitations of our study.

Finally, we have added the following sentence in discussion section, paragraph 5, lines 274 – 296.

On the other hand, the A/A and T/A genotypes of the CD147 rs8259T>A variant have been associated with higher expression levels of CD147 mRNA or protein in PBMCs from patients with ACS but not in patients with stable angina (23) Thus, the effect of genotypes of the CD147 rs8259T>A variant on the expression of its mRNA seems to de-pend on the cell type, tissue, or disease. As far as we know, the expression of miR-492 had not been evaluated in patients with COVID-19, but a study showed that this miRNA is expressed in PBMCs from healthy controls (24). In line with this, we also found an expression of this miRNA in whole blood cells of our controls. miR-492 has been report-ed to bind to the T allele of the CD147 rs8259T>A variant, meanwhile, the A allele destroys the binding site for this miRNA, which generates an increase in CD147 protein expression (24). In this same study was reported that the AA genotype of rs8259T>A is associated with increased production of CD147 protein (but not mRNA) in PBMCs from patients with psoriasis versus the TT genotype (24). Therefore, the effect of the three genotypes (T/T, T/A, and A/A) of this variant on its protein or mRNA expression levels appears to depend on cell type, tissue, or disease (23,24). In line with this, we report a finding like that reported by Wu et al., where they do not observe statistically significant differences in CD147 mRNA levels taking into account the genotypes of this variant in patients with psoriasis and stable angina (23,24). Although we did not identify differences in TNF mRNA expression considering the three rs8259T>A genotypes, we know that TNF is a cytokine that mediates inflammation and can cause detrimental tissue damage and promotes lung fibrosis, which later results in pneumonia, pulmonary edema, acute respiratory distress syndrome, promote inflammation and be associated with morbidity and mortality in patients with COVID-19 (25).

Reviewer 2 Report

The study conducted by Amezcua-Guerra et al. involved genotyping the CD147 rs8259 variant in a sample of 195 COVID-19 patients and 185 healthy controls from the Mexican population. The authors identified the A allele and AA genotype of the CD147 rs8259 variant as risk factors for COVID-19 in the Mexican population, suggesting the potential involvement of CD147 in the pathogenesis of the disease. Understanding the role of CD147 and its variants in COVID-19 susceptibility may contribute to personalized medicine and the development of targeted therapeutic interventions. However, there are several points that should be considered:

1.     As indicated by the authors, the study included a relatively small sample size of 195 COVID-19 patients and 185 healthy controls. A larger sample size would provide more robust and reliable results.

2.     The study focused solely on individuals from the Mexican population, limiting the generalizability of the findings to other ethnicities or regions. Further studies involving diverse populations are necessary to validate the role of the CD147 rs8259 variant in COVID-19 susceptibility.

3.     The study primarily focused on the association between the CD147 rs8259 variant and COVID-19 susceptibility without delving into the functional implications of the variant. Conducting functional studies would be beneficial to understand the underlying mechanisms and biological relevance of this variant in the context of COVID-19.

4.     The study did not account for potential confounding factors such as age, comorbidities, or other genetic variations that could influence COVID-19 susceptibility. Controlling for these factors is important to ensure that the observed associations are specific to the CD147 rs8259 variant.

5.     While the study identified an association between the CD147 rs8259 variant and COVID-19 susceptibility in the Mexican population, independent replication studies are needed to validate these findings. Replication in different populations and cohorts would strengthen the evidence for this association.

6.     The study did not provide any functional genomics data to support the hypothesis of how the CD147 rs8259 variant affects COVID-19 susceptibility. Including functional genomics analyses, such as gene expression profiling or protein analysis, could provide additional insights into the molecular mechanisms involved.

Typos and grammer issues

Author Response

Reviewer 2

The study conducted by Amezcua-Guerra et al. involved genotyping the CD147 rs8259 variant in a sample of 195 COVID-19 patients and 185 healthy controls from the Mexican population. The authors identified the A allele and AA genotype of the CD147 rs8259 variant as risk factors for COVID-19 in the Mexican population, suggesting the potential involvement of CD147 in the pathogenesis of the disease. Understanding the role of CD147 and its variants in COVID-19 susceptibility may contribute to personalized medicine and the development of targeted therapeutic interventions. However, there are several points that should be considered:

Question 1.     As indicated by the authors, the study included a relatively small sample size of 195 COVID-19 patients and 185 healthy controls. A larger sample size would provide more robust and reliable results.

Answer 1. Dear reviewer, we agree with your comment. Therefore, we have added some sentences that will not be so conclusive. Some of the phrases that we have added are the following:

Abstract section, lines 30 – 31:

Our results suggest that the CD147 rs8259T>A variant is a risk factor for COVID-19.

Introduction section, paragraph 2, lines 63 – 73:

Izmailova and colleagues aimed to investigate potential associations between the CD147 rs8259T>A, ACE2 rs4240157T>C, TMPRSS2 rs12329760C>T, and TMPRSS11A rs353163C/T variants and COVID-19 severity in the Ukrainian population. They analyzed these variants among cases divided into three groups: without oxygen therapy, non-invasive oxygen therapy, and invasive oxygen therapy. Interestingly, they found frequency differences for the TMPRSS2 rs12329760C>T variant only in the group receiving invasive oxygen therapy but they did not identify an association between CD147 rs8259T>A and COVID-19 susceptibility or severity (12). However, given the sample size reported in that study, it is necessary to evaluate whether this same variant is a risk factor in populations with different genetic background.

In discussion section (paragraph 2, lines 226 – 233), we have also added the following sentence:

To our knowledge, only one study has reported the role of the rs8259T>A variant in the susceptibility and severity of COVID-19 in the Ukrainian population, but it was not as-sociated with COVID-19 susceptibility or severity (12). Our findings differ from those reported in the Ukrainian population because we identified an association with COVID-19 susceptibility, meanwhile, they did not report any association. Some differences between the two studies might explain this discrepancy. Firstly, we had a larger number of control samples, including 185 Mexican mestizos compared to their 92 controls.

Conclusion section, lines 302 - 303:

Our data suggests that this CD147 rs8259T>A variant is a risk factors for COVID-19 in the Mexican population.

Question 2. The study focused solely on individuals from the Mexican population, limiting the generalizability of the findings to other ethnicities or regions. Further studies involving diverse populations are necessary to validate the role of the CD147 rs8259 variant in COVID-19 susceptibility.

Answer 2. We agree with your comment. Therefore, we have added the following sentences:

Discussion section, paragraph 2, lines 226 – 244:

To the best of our knowledge, only one study has reported the role of the rs8259T>A variant in the susceptibility and severity of COVID-19 in the Ukrainian population, but it was not associated with COVID-19 susceptibility or severity (12). Our findings differ from those reported in the Ukrainian population because we identified an association with COVID-19 susceptibility, meanwhile, they did not report any association. Some differences between the two studies might explain this discrepancy. Firstly, we had a larger number of control samples, including 185 Mexican mestizos compared to their 92 controls. Ancestry also plays a crucial role, our study includes participants from Central Mexico, which is characterized by approximately 52% European, 44% Amerindian, and 4% African ancestry (20). In fact, genotype frequencies of TT, TA, and AA of the CD147 rs8259T>A variant in our Mexican controls is different from those identified in Ukrainian controls (12). These differences could lead to the fact that in Mexicans it is a risk fac-tor, while in Ukrainians no. Other studies (considering the lack of association reported in the European population and the association we identified in a Latin American population) should be conducted in other populations with different genetic background to determine whether this CD147 variant is indeed a risk factor for SARS-CoV-2 infection and COVID-19.

Question 3.  The study primarily focused on the association between the CD147 rs8259 variant and COVID-19 susceptibility without delving into the functional implications of the variant. Conducting functional studies would be beneficial to understand the underlying mechanisms and biological relevance of this variant in the context of COVID-19.

Answer 3. We appreciate your comment. However, we did not conduct any functional study of the three genotypes of the CD147 rs8259T>A variant to determine whether it affects the expression level of its mRNA and protein or to determine whether this variant influences the miR-492 binding. There are two studies showing the effect of this CD147 variant on CD147 mRNA and protein expression and on miR-492 binding, thus, both studies are mentioned in detail in our manuscript.

Discussion section, paragraph 5, lines 274 - 296

On the other hand, the A/A and T/A genotypes of the CD147 rs8259T>A variant have been associated with higher expression levels of CD147 mRNA or protein in PBMCs from patients with ACS but not in patients with stable angina (23) Thus, the effect of genotypes of the CD147 rs8259T>A variant on the expression of its mRNA seems to de-pend on the cell type, tissue, or disease. As far as we know, the expression of miR-492 had not been evaluated in patients with COVID-19, but a study showed that this miRNA is expressed in PBMCs from healthy controls (24). In line with this, we also found an expression of this miRNA in whole blood cells of our controls. miR-492 has been report-ed to bind to the T allele of the CD147 rs8259T>A variant, meanwhile, the A allele destroys the binding site for this miRNA, which generates an increase in CD147 protein expression (24). In this same study was reported that the AA genotype of rs8259T>A is associated with increased production of CD147 protein (but not mRNA) in PBMCs from patients with psoriasis versus the TT genotype (24). Therefore, the effect of the three genotypes (T/T, T/A, and A/A) of this variant on its protein or mRNA expression levels appears to depend on cell type, tissue, or disease (23,24). In line with this, we report a finding like that reported by Wu et al., where they do not observe statistically significant differences in CD147 mRNA levels taking into account the genotypes of this variant in patients with psoriasis and stable angina (23,24). Although we did not identify differences in TNF mRNA expression considering the three rs8259T>A genotypes, we know that TNF is a cytokine that mediates inflammation and can cause detrimental tissue damage and promotes lung fibrosis, which later results in pneumonia, pulmonary edema, acute respiratory distress syndrome, promote inflammation and be associated with morbidity and mortality in patients with COVID-19 (25).

Question 4.     The study did not account for potential confounding factors such as age, comorbidities, or other genetic variations that could influence COVID-19 susceptibility. Controlling for these factors is important to ensure that the observed associations are specific to the CD147 rs8259 variant.

Answer 4. We have done this type of analysis. However, we had not placed any sentence regarding the confounding factors, now we have added the following sentence in the results section (Results section, paragraph 2, lines 162 - 165).

Our analyses revealed after adjusting for age, gender, comorbidities, etc., an association between the CD147 rs8259T>A SNV and COVID-19 infection; T vs A (OR 1.36, 95% CI 1.02 to 1.81, and p = 0.037) and TT vs AA (OR 1.77, 95% CI 1.01 to 3.09 and p = 0.046) (Figure 2).

  1. While the study identified an association between the CD147 rs8259 variant and COVID-19 susceptibility in the Mexican population, independent replication studies are needed to validate these findings. Replication in different populations and cohorts would strengthen the evidence for this association.

Answer 5. We agree with your suggestion, so we have added the following sentence in the introduction section, paragraph 2, lines 63 - 73.

Izmailova and colleagues aimed to investigate potential associations between the CD147 rs8259T>A, ACE2 rs4240157T>C, TMPRSS2 rs12329760C>T, and TMPRSS11A rs353163C/T variants and COVID-19 severity in the Ukrainian population. They analyzed these variants among cases divided into three groups: without oxygen therapy, non-invasive oxygen therapy, and invasive oxygen therapy. Interestingly, they found frequency differences for the TMPRSS2 rs12329760C>T variant only in the group receiving invasive oxygen therapy but they did not identify an association between CD147 rs8259T>A and COVID-19 susceptibility or severity (12). However, given the sample size reported in that study, it is necessary to evaluate whether this same variant is a risk factor in populations with different genetic background.

We have also added the following sentence in discussion section, paragraph 2, lines 226 – 244.

To the best of our knowledge, only one study has reported the role of the rs8259T>A variant in the susceptibility and severity of COVID-19 in the Ukrainian population, but it was not as-sociated with COVID-19 susceptibility or severity (12). Our findings differ from those reported in the Ukrainian population because we identified an association with COVID-19 susceptibility, meanwhile, they did not report any association. Some differences between the two studies might explain this discrepancy. Firstly, we had a larger number of control samples, including 185 Mexican mestizos compared to their 92 controls. Ancestry also plays a crucial role, our study includes participants from Central Mexico, which is characterized by approximately 52% European, 44% Amerindian, and 4% African ancestry (20). In fact, genotype frequencies of TT, TA, and AA of the CD147 rs8259T>A variant in our Mexican controls is different from those identified in Ukrainian controls (12). These differences could lead to the fact that in Mexicans it is a risk fac-tor, while in Ukrainians no. Other studies (considering the lack of association reported in the European population and the association we identified in a Latin American population) should be conducted in other populations with different genetic background to determine whether this CD147 variant is indeed a risk factor for SARS-CoV-2 infection and COVID-19

Question 6.     The study did not provide any functional genomics data to support the hypothesis of how the CD147 rs8259 variant affects COVID-19 susceptibility. Including functional genomics analyses, such as gene expression profiling or protein analysis, could provide additional insights into the molecular mechanisms involved.

Answer 6. We agree with your comment, however, the study design proposed from the beginning in our study was focused on evaluating whether this variant is a risk or severity factor for COVID-19, our results have shown that it is a risk factor for the development of COVID-19 but is associated with severity. We also wanted to assess whether CD147 levels are higher in patients than in controls, finding an association. Finally, we wanted to replicate the data on the effect of the three genotypes of this variant on the mRNA levels of CD147, TNF, and miR-492, which was not replicated in our study; part of the explanation could be because it has been observed that the effect of these three genotypes seems to depend on the type of cells, tissues, or disease. Therefore, it would be interesting to evaluate a profile of mRNAs or proteins considering the genotypes of these variants, since it would provide us with additional information on the effect of this variant. However, this goal is beyond our reach. Thus, we have respectfully decided not to add any sentences regarding this in our manuscript.